# Prognostic and Diagnostic Power of Delta Neutrophil Index and Mean Platelet Component in Febrile Patients with Suspected Sepsis

**DOI:** 10.3390/biomedicines11123190

**Published:** 2023-11-30

**Authors:** Taehun Lee, Jongwook Lee, Dong Hoon Shin, Hyungdon Lee, Soo-Ki Kim

**Affiliations:** 1Department of Emergency Medicine, College of Medicine, Hallym University, Chuncheon Sacred Heart Hospital, Chuncheon 24253, Republic of Korea; ion2674@naver.com; 2Department of Laboratory Medicine, Konyang University Hospital, Daejeon 35465, Republic of Korea; lee423619@hanmail.net; 3Department of Laboratory Medicine, College of Medicine, Hallym University, Chuncheon Sacred Heart Hospital, Chuncheon 24253, Republic of Korea; shindh@hallym.or.kr; 4Department of Internal Medicine, College of Medicine, Hallym University, Chuncheon Sacred Heart Hospital, Chuncheon 24253, Republic of Korea; 5Department of Microbiology, Wonju College of Medicine, Research Institute of Metabolism and Inflammation Research, Yonsei University, Wonju 26426, Republic of Korea

**Keywords:** delta neutrophil index, sepsis, mean platelet component, procalcitonin, CRP, emergency department, fever

## Abstract

Background: The delta neutrophil index (DNI), a prognostic and diagnostic marker for sepsis, is based on the leukocyte count. Platelet activation, similar to leukocyte activation, plays a crucial role in host defense against pathogens and may serve as a predictor of sepsis outcome. However, the combined evaluation of mean platelet component (MPC) and DNI has rarely been used to assess sepsis. Methods: To assess the prognostic and diagnostic validity of the simultaneous evaluation of DNI and MPC in cases of human febrile sepsis, we conducted measurements of cellular indices, including DNI and MPC, as well as molecular biomarkers, including procalcitonin (PCT) and C-reactive protein (CRP). This study was carried out in patients admitted to the emergency department with suspected sepsis. Results: Using a cutoff value of 2.65%, the sensitivity, specificity, positive predictive value (PPV), and negative predictive value (NPV) of the DNI in sepsis were found to be 69%, 73.9%, 77.9%, and 64.1%, respectively. Furthermore, significant differences in DNI and MPC levels were observed between the sepsis and non-sepsis groups (6.7 ± 7.8% versus 2.1 ± 2.2% (*p* = 0.000) and 26.0 ± 1.9 g/dL versus 26.8 ± 1.4 g/dL (*p* = 0.002), respectively). Notably, there was a negative correlation between DNI and MPC, with the strength of the correlation varying based on the cause of sepsis. By setting the cutoff value of the DNI to 6.2%, its sensitivity, specificity, and NPV improved to 100%, 80.3%, and 100%, respectively, although the PPV remained at 10.6%. Conclusions: In our study, the DNI demonstrates superior effectiveness compared with other molecular biomarkers, such as CRP and procalcitonin, in distinguishing septic febrile patients from non-septic febrile patients. Additionally, a negative correlation exists between MPC and DNI, making MPC a valuable marker for differentiating the etiology of sepsis. These findings hold significant clinical implications, as DNI/MPC evaluation is a cost-effective and readily applicable approach in various impending sepsis scenarios. Notably, this study represents the first examination of the prognostic and diagnostic validity of employing the simultaneous evaluation of DNI and MPC in human cases of febrile sepsis.

## 1. Introduction

Fever in humans is characterized by an oral temperature of 37.7 °C or higher, as measured with a thermometer [1]. Various factors can contribute to fever, including infection [2,3], trauma [4], metabolic disease [5], and autoimmune disease [6,7]. Both infectious and non-infectious causes can result in systemic inflammatory response syndrome (SIRS). However, SIRS associated with infection tends to have higher mortality rates than SIRS caused by non-infectious conditions [8]. Consequently, infection-related fever is recognized as a predictor of sepsis development. Molecular and cellular biomarkers can aid in differentiating sepsis from SIRS, even in the presence of fever [9]. Cellular biomarkers, in particular, are preferred due to their ease of use, immediate application, high reproducibility, and cost-effectiveness [10].

Previous studies [11] have shown that a complete blood count can identify early risk of developing sepsis and ascertain a poor prognosis. Specifically, total white blood cells are composed of lymphocytes, monocytes, neutrophils, eosinophils, and basophils, and their increase is known to be caused by inflammation or infection; however, their increase can also be caused by non-infectious diseases, such as rheumatoid arthritis or cancer, so it is known to have low specificity in sepsis. Neutrophils are the most abundant in peripheral blood and quickly migrate to the site of infection to remove infectious agents in the event of infection; their levels are increased in sepsis, but these increases can be caused by smoking or stress, as well as inflammatory bowel disease or rheumatic diseases, so they are not suitable for the diagnosis and prognosis of sepsis. Eosinophils also migrate to the site of infection and secrete cytokines; eosinophil reduction has shown good sensitivity and specificity in the diagnosis of sepsis in some studies [12], but in a recent analysis [13], it was not found to be superior to CRP or PCT. Furthermore, lymphocytes are important components of the adaptive immune response, and although lymphopenia is associated with poor prognosis in sepsis, it can also be found in nutritional deficiencies and autoimmune diseases [14]. On the other hand, the ratio of neutrophils to lymphocytes reflects the diagnosis and prognosis of sepsis, and its evaluation is known to be superior to that of CRP and inferior to that of PCT [15]; however, this ratio is also increased in coronary artery disease and cancer, limiting its use [16].

Previous studies [17] have shown that platelet dysfunction is present in sepsis and associated with a poor prognosis and that it includes thrombocytopenia, mean platelet volume, and immature platelet fraction. Of these, thrombocytopenia was associated with higher risk of death in the intensive care unit [18]. However, thrombocytopenia is also associated with non-infectious diseases, such as idiopathic thrombocytopenic purpura and systemic lupus erythematosus. Mean platelet volume is higher in sepsis than in localized infections, and an increased mean platelet volume has been reported to be associated with more severe infections or antibiotic treatment failure; however, it is also increased in heart disease and cerebral infarction, limiting its applicability to sepsis.

In sepsis, an important characteristic leukocyte change is the appearance of immature neutrophils [19]. Building upon this observation, the delta neutrophil index (DNI) has shown promise as a cellular biomarker for early sepsis differentiation, as demonstrated in our preclinical study [20] and other clinical reports [21,22] on the early differentiation of sepsis. In acute severe sepsis or septic shock, the DNI peaks at the time of onset and gradually decreases 24 h after onset [23], and in non-infectious inflammation such as severe acute pancreatitis, DNI values are also observed to increase and then decrease in milder cases [24]. Similarly, platelet activation in sepsis plays a crucial role in host defense against pathogens and holds predictive value for sepsis outcomes. Mean platelet component (MPC) is a recognized cellular biomarker for sepsis [25]. Fortunately, automated blood analyzers can readily measure both leukocyte indices, like the DNI, and platelet indices, such as MPC [26]. Although both DNI and MPC have demonstrated prognostic capabilities for sepsis, there has been limited evaluation of their simultaneous predictive potential. We hypothesized that assessing DNI and MPC together in patients with suspected sepsis can provide a more robust and accurate diagnosis of sepsis and prognostic prediction than evaluating DNI or MPC alone, as well as other biomarkers, like CRP and procalcitonin. To investigate this, we compared the cellular markers DNI and MPC, and the molecular biomarkers procalcitonin and C-reactive protein (CRP) in patients presenting with suspected sepsis.

## 2. Material and Methods

### 2.1. Sample Size Calculation

To determine the sample size of patients with suspected sepsis, we utilized a significance level (type I error) of 0.1% and a type II error of 0.2%. Previous studies [27,28] reported the incidence of confirmed sepsis among patients with suspected sepsis to range from 0.39 to 0.43. Additionally, we assumed that the control group had to consist of 1.5 times more individuals than the confirmed-sepsis group for patients with suspected sepsis. Using the appropriate formula, the calculated total sample size was determined to be 180. Considering a dropout rate of 10%, the final total sample size was determined to be 198.

q_1_ = proportion of septic febrile patients in febrile patients with suspected sepsis.q_2_ = proportion of non-febrile septic patients in non-febrile patients with suspected sepsis.P_1_ = proportion of patients expected to have febrile sepsis in febrile patients with suspected sepsis.P_2_ = proportion of patients expected to have non-febrile sepsis in non-febrile patients with suspected sepsis.Z_α_ = standard normal deviate for α.Z_β_ = standard normal deviate for β.N = total number of febrile patients with suspected sepsis.P = q_1_ P_1_ + q_2_ P_2._

Then,
N = [Z_α_ √P(1 − P)(1/q_1_ + 1/q_2_) + Z_β_ √P_1_(1 − P_1_)(1/q_1_) + P_2_(1 − P_2_)(1/q_2_)]^2^/(P_1_ − P_2_)^2^

### 2.2. Study Population

This study was carried out at Chuncheon Sacred Heart Hospital, a 400-bed hospital located in Chuncheon, South Korea. The study was approved by the Institutional Review Board of Chuncheon Sacred Heart Hospital (IRB No. 2021-04-001). As this study was retrospectively designed, informed consent was not obtained from the Institutional Review Board. We subsequently enrolled all febrile patients who visited the emergency department between January 2021 and March 2021. 

Inclusion criteria for patients with suspected sepsis:-Body temperature > 38 °C;-Patient’s age ≥ 19 years.

Exclusion criterion for patients with suspected sepsis:-Age < 19 years old.

### 2.3. Data Definitions; Data Collection; and Definitions of Infection, Sepsis, Suspected Sepsis, and Organ Failure

Basic demographic data of patients, along with laboratory results including PCT, CRP, MPC, hemoglobin level, blood urea nitrogen, serum creatinine, leukocyte count, and platelet count, were collected from electronic medical records (EMRs). The highest recorded temperature on the day of the emergency department visit was noted. Clinical variables such as mortality, organ failure, underlying disease, admission to the intensive care unit (ICU), hospital days, surgeries, and readmissions within 30 days of discharge, as well as culture results from blood, urine, sputum, and wounds, were also extracted from the EMRs.

Sepsis was defined as the presence of both infection and systemic inflammatory response syndrome [29]. Suspected sepsis was defined as a patient admitted to the emergency department with a fever of 38 °C or higher. Infection was defined as the identification of causative organisms through culture of fluids collected from sites of infection, including the lung, kidney, skin, bone, soft tissue, and liver. The criteria for systemic inflammatory response syndrome were defined based on a previous study [28]. Organ failure was defined in accordance with sequential organ failure assessment [30]. 

### 2.4. DNI Calculation and Other Blood Test Measurements

Blood samples for DNI measurement were collected in ethylenediaminetetraacetic acid (EDTA) tubes. DNI measurements were performed within 1 min of blood collection using an automatic cell analyzer (ADVIA 2120 Hematology System; Siemens Healthcare Diagnostics, Forchheim, Germany). The calculation of the DNI involves two independent white blood cell (WBC) analysis methods: the MPO channel and the lobularity/nuclear density channel. The DNI calculation formula incorporates the neutrophil and eosinophil subfractions from the MPO channel, as well as the polymorphonuclear neutrophil (PMN) subfraction from the nuclear lobularity channel.

### 2.5. Statistical Analysis 

We used the Kolmogorov–Smirnov test to determine the distribution of continuous variables, which are presented as means ± standard deviations (SDs), Student’s *t*-test for normally distributed variables, and the Mann–Whitney U test for non-normally distributed variables. Categorical variables are presented as percentages and compared using the chi-square test or Fisher’s exact test [31].

The Pearson rank correlation coefficient was used to evaluate the associations among DNI, MPC, PCT, and CRP. Area under the curve (AUC) values were used to distinguish the DNI from the other markers. Receiver-operating characteristic (ROC) curves were constructed, and the Youden’s index was applied to measure sensitivity and specificity at optimal cutoff values of the DNI. A *p*-value lower than 0.05 was considered statistically significant. All statistical analyses were performed using IBM SPSS statistics for Windows, version 23.0 (IBM Corp., Armonk, NY, USA), and R studio, version 4.3 (Rstudio, Inc., Boston, MA, USA).

## 3. Results

A total of 215 patients were enrolled in the study. We categorized the causes of septic and non-septic cases (Table 1). 

In the non-sepsis group, the gastrointestinal cause was enteritis in more than 90% of cases, and enteritis was diagnosed according to the presence of symptoms such as diarrhea and abdominal pain but no fever of other causes in computed tomography, or evidence of severe infection in hematology tests. The upper respiratory tract cause was diagnosed according to the absence of lower respiratory tract infection, such as no lung infiltrates being found in chest X-ray and no rales in breath sounds. Unexplained fever was diagnosed in the absence of severe infection in blood tests and where there was no evidence of bacteria in X-ray or culture. Musculoskeletal and soft tissue cases were diagnosed when there was tissue inflammation or redness but no infection. The basic clinical characteristics of the patients included in the study are shown in Table 2. 

CRP, procalcitonin, and DNI levels in the sepsis group were higher than those in the non-sepsis group (103.0 ± 91.6 vs. 34.9 ± 45.7 (*p* = 0.000), 8.2 ± 21.1 vs. 1.8 ± 6.8 (*p* = 0.004), and 6.7 ± 7.8 vs. 2.1 ± 2.2 (*p* = 0.000)). When the cutoff value of the DNI was 2.65%, its sensitivity, specificity, positive predictive value (PPV), and negative predictive value (NPV) in sepsis were 69%, 73.9%, 77.9%, and 64.1%, respectively. A comparison of the accuracy of sepsis outcome resulting from assessments of DNI, CRP, and procalcitonin is shown in Figure 1. 

Using cutoff values of 6.3% for the DNI, 0.48 ng/mL for procalcitonin, and 172.4 mg/dL for CRP, the AUC values were 0.844, 0.843, and 0.835 (*p* < 0.001), respectively, with significant differences, and the DNI had the highest AUC value (Figure 1). The sepsis group had a lower MPC level than the non-sepsis group (26.0 ± 1.9 vs. 26.8 ± 1.4, *p* = 0.002). The sepsis group had a longer total length of stay than the non-sepsis group (8.6 ± 7.4 vs. 4.0 ± 6.6, *p* = 0.000). The ICU group also had a higher DNI than the non-ICU group (8.7 ± 10.8 vs. 3.9 ± 4.9, *p* = 0.015). In addition, the MPC level in the ICU group was significantly lower than that in the non-ICU group (25.1 ± 2.3 vs. 26.6 ± 1.5, *p* = 0.001). Looking at the effect of the source of sepsis, the MPC level was the lowest in sepsis that originated in the lungs (Table 3).

Table 4 represents the Pearson rank correlation coefficients of the DNI with other inflammatory markers, including MPC, CRP, and PCT, categorized by cause of sepsis. In all three categories (all causes, gastrointestinal cause, and pulmonary cause), the DNI exhibits negative correlations with MPC.

The ROC curves illustrated in Figure 1 demonstrate that the DNI is the most effective factor for predicting mortality in sepsis, with an area under the curve (AUC) of 0.88 (95% confidence interval (CI), 0.81–0.95).

### Differentiating DNI and Other Markers Associated with Mortality in Septic Patients

Using a cutoff level of 6.2% for the DNI, this marker was found to be superior in predicting sepsis-related mortality compared with other markers, such as CRP and procalcitonin.

The sensitivity, specificity, positive predictive value (PPV), and negative predictive value (NPV) of the DNI in mortality prediction were 100%, 80.3%, 10.9%, and 100%, respectively (as shown in Table 5).

## 4. Discussion

The present study aimed to assess the effectiveness of the DNI compared with other molecular biomarkers, i.e., PCT and CRP, in differentiating septic febrile patients from non-septic febrile patients in the emergency department and predicting prognosis. In the early stages of sepsis, immature neutrophils are present in peripheral blood to compensate for the lack of mature neutrophils. The number of neutrophil bands indicates the number of immature neutrophils and is increased in sepsis. The DNI is expressed as the fraction of immature neutrophils in the total number of neutrophils; this is assessed using automated analysis machines and is obtained by calculating the difference between the nuclear lobularity channel and the MPO channel [32,33]. It is worth noting that previous studies [34] have established the utility of procalcitonin as a biomarker for diagnosing sepsis and predicting the prognosis. However, our study demonstrates that the DNI is a superior cytological marker for diagnosing and predicting sepsis when compared with procalcitonin.

Furthermore, platelet assessment is known to be clinically useful in the diagnosis and prognosis of sepsis [11]. Activated platelets that adhere, aggregate, and degranulate in injured tissue express increased CD62P, which is known as platelet activation marker; platelets are activated by histamine, oxygen radicals, interleukin 1(IL-1), and tumor necrosis factor (TNF), and cytokines such as IL-1 and TNF are increased in sepsis [35,36]. Mean platelet component (MPC), which is reflected by CD62P levels and is calculated as mean platelet density (100 × platelet mass(pg)/platelet volume), is decreased during platelet activation [37]. Recently, platelet indices, such as MPC, that are available on automated hematology analyzers have shown to be beneficial in differentiating sepsis [38]. Furthermore, our findings reveal a negative correlation between MPC levels and DNI levels, indicating the potential of MPC in differentiating the causes of sepsis. These conclusions are drawn from the statistical analysis and comparative measurements of two molecular markers (PCT and CRP) and two cellular markers (DNI and MPC).

In our study, DNI levels increased above the baseline (1%) in both the sepsis and non-sepsis febrile groups. However, this increase was more pronounced in the sepsis febrile group than in the non-sepsis febrile group (110 out of 123 (89.4%) vs. 61 out of 92 (66.3%), *p* = 0.000). There was a significant difference in DNI levels between the sepsis and non-sepsis febrile groups (6.7 ± 7.8% vs. 2.1 ± 2.2%, *p* = 0.000).

The elevated DNI levels observed in the non-sepsis febrile group may be attributed to non-infectious inflammation, such as trauma-related tissue damage, metabolic disorders, or autoimmune diseases. However, DNI levels were significantly higher in the sepsis febrile group than in the non-sepsis febrile group, which supports previous reports highlighting the effectiveness of the DNI in assessing sepsis [39]. Accumulative studies have also suggested that the DNI could be a diagnostic marker for sepsis [29,30,40,41].

Consistent with prior research, our results demonstrate that the DNI is effective in differentiating sepsis-related fever from non-sepsis-related fever. Furthermore, the DNI with the cutoff value of 6.2% was more effective than CRP and PCT in predicting the prognosis of febrile sepsis, including outcomes such as mortality, organ failure, length of stay, and in-hospital surgery. Patients with a DNI value greater than 6.2% had a worse prognosis.

It is well-known that platelets are activated in various infectious and non-infectious pathological conditions, such as acute myocardial infarction [42], cerebrovascular disease [43], and diabetes mellitus [44]. Similar to the aberrant neutrophil behavior observed in sepsis, platelet activation plays a significant role in these conditions.

Patients with febrile sepsis and lower MPC levels are at increased risk of death. This association can be attributed to the detrimental effects of platelet activation, including platelet depletion, thrombosis, and the development of disseminated intravascular coagulopathy [45]. These complications can ultimately lead to multi-organ failure and even death [46].

These findings have significant clinical implications, suggesting that patients admitted to the emergency department with fever and suspected sepsis, particularly those with low MPC levels, should undergo prompt evaluation to identify the source of sepsis and initiate appropriate management strategies as early as possible. 

Finally, inferential statistics revealed a significant negative correlation between DNI and MPC, and the strength of this inverse correlation varied depending on the origin of sepsis. Notably, this is the first report to demonstrate this relationship in the context of human sepsis. The strongest negative correlation between DNI and MPC was observed in cases of sepsis originating from the gastrointestinal tract and lungs (Pearson correlation coefficients: −0.611 for gastrointestinal origin, −0.492 for pulmonary origin, and −0.312 for all sources of sepsis). This finding may provide valuable insights into the clinical application of identifying the focus of infection in sepsis. However, the use of a panel of the best candidate markers may be superior to the use of CRP, procalcitonin, or DNI alone in assessing the diagnosis and prognosis of sepsis, but there is not much room for consideration at this time, because each biomarker alone is excellent for the diagnosis and prognosis of sepsis.

One possible explanation for the strong negative correlation between DNI and MPC in gastrointestinal and pulmonary sepsis could be the dysbiosis of gut microbiota associated with the gut–lung axis [47]. The human gut harbors a vast number of microbiomes, accounting for over 80% of the body’s lymphocytes, and plays a crucial role in the development and function of immune cells [48]. Under septic conditions, the gut microbiota undergoes dysbiosis [49] and increased permeability, leading to platelet activation, severe tissue damage, and ultimately multi-organ failure and death [50]. Similarly, the gut–lung axis (GLA) can significantly impact the lungs during sepsis [51]. Gut microbiota can modulate pulmonary immune responses and diseases through the mesenteric lymphatic system [48,52]. Therefore, we speculate that these intricate networks contribute to the more pronounced negative correlations observed in gastrointestinal- and pulmonary-origin sepsis compared with other causes of sepsis.

Our study demonstrates the predictive value of DNI and MPC in febrile sepsis and highlights the potential benefit of utilizing both markers together to enhance the probability of predicting sepsis. Nevertheless, it is important to acknowledge the limitations of this study. Firstly, there were cases where initial cultures were not obtained in fewer than 5% of patients. Additionally, follow-up data were unavailable for fewer than 10% of eligible patients. Furthermore, patients with sepsis without fever were not evaluated. The one-year follow-up period in this study may not provide a comprehensive understanding of the long-term prognosis and outcome assessment of febrile patients with suspected sepsis. Therefore, future research with longer follow-up durations, ideally one year or more, is warranted. Lastly, it is important to note that this study utilized a retrospective design.

## 5. Summary

This study highlights the superiority of the DNI over other molecular biomarkers, such as procalcitonin and CRP, in differentiating septic febrile patients from non-septic febrile patients in the emergency department. MPC also contributes to the differentiation of causes in febrile sepsis. The clinical significance of DNI/MPC lies in their cost-effectiveness and applicability in various impending sepsis scenarios. Importantly, this is the first study to evaluate the diagnostic validity of simultaneously employing DNI and MPC in human cases of febrile sepsis.

## Figures and Tables

**Figure 1 biomedicines-11-03190-f001:**
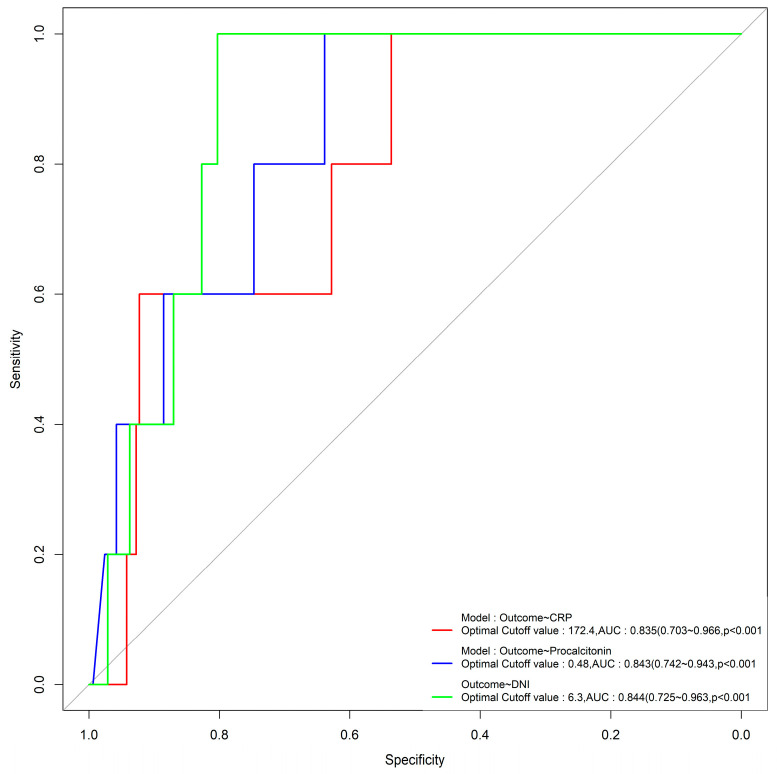
ROC curves of DNI, CRP, and procalcitonin in relation to sepsis.

**Table 1 biomedicines-11-03190-t001:** Causes of septic and non-septic febrile patients.

Origin	Number (%)	DNI (%)
Sepsis		
Urinary tract infection	49 (22.8)	6.6 ± 9.1
Respiratory tract infection	30 (14.0)	5.7 ± 5.9
Gastrointestinal tract infection	31 (14.4)	8.7 ± 8.4
Soft tissue infection	11 (5.1)	3.6 ± 1.9
Central nervous system infection	2 (0.9)	5.5 ± 7.3
Not sepsis	92 (42.8)	2.1 ± 2.2
Gastrointestinal tract infection	39 (42.9)	
Cancer	10 (11.0)	
Fever of unknown origin	10 (11.0)	
Upper respiratory tract	9 (9.9)	
Musculoskeletal system	5 (5.5)	
Urinary tract infection	4 (4.4)	
Soft tissue infection	4 (4.4)	
Central nervous system infection	3 (3.3)	
Drug intoxication	2 (2.2)	
Rheumatic disease	1 (1.1)	
Hyperventilation	1 (1.1)	
Heart failure	1 (1.1)	
Transfusion	1 (1.1)	
Others	2 (2.2)	
Total	215 (100)	4.7 ± 6.4

**Table 2 biomedicines-11-03190-t002:** Clinical characteristics of patients in sepsis and non-sepsis groups.

Characteristic	Sepsis Group	Non-Sepsis Group	*p*-Value
Number (count)	123	92	
Sex (Male (%))	58 (47.2)	40 (43.5)	0.592
Age (years)	63.2 ± 19.3	50.7 ± 24.0	0.000
ICU admission (n (%))	27 (22.0)	8 (8.7)	0.009
Multi-organ failure (n (%))	9 (7.4)	0 (0)	0.011
Organ failure (n (%))	34 (27.6)	7 (7.6)	0.000
CRP (mg/L)	103.0 ± 91.6	34.9 ± 45.7	0.000
Procalcitonin (ng/mL)	8.2 ± 21.1	1.8 ± 6.8	0.004
DNI (%)	6.7 ± 7.8	2.1 ± 2.2	0.000
MPC (g/dL)	26.0 ± 1.9	26.8 ± 1.4	0.002
Total admission (days)	8.6 ± 7.4	4.0 ± 6.6	0.000
ICU admission (days)	1.4 ± 3.3	0.6 ± 2.8	0.071
Mortality (n (%))	5 (4.1)	0	0.073
Stroke (n (%))	3 (3.3)	3 (2.4)	1.000

ICU, intensive care unit; CRP, C-reactive protein; DNI, delta neutrophil index; MPC, mean platelet component.

**Table 3 biomedicines-11-03190-t003:** MPC levels in febrile patients by origin of sepsis.

Origin of Sepsis	Number (Count)	Average ± Standard Deviation	*p*-Value
Pulmonary	30	25.2 ± 2.3	
Gastrointestinal	31	25.9 ± 2.3	
Soft tissue	11	26.1 ± 2.0	
Urinary	49	26.4 ± 1.4	
Central nervous system	2	28.3 ± 1.8	
Non-septic	92	26.8 ± 1.4	
Total	215	26.3 ± 1.7	0.000

ICU, intensive care unit; CRP, C-reactive protein; DNI, delta neutrophil index; MPC, mean platelet component.

**Table 4 biomedicines-11-03190-t004:** Pearson rank correlation coefficients of DNI with other inflammatory markers by cause of sepsis.

Origin of Sepsis		DNI	MPC	CRP	PCT
All septic causes	DNI Pearson correlation	1	−0.312	0.047	0.053
	*p*-value (two-sided)		0.000	0.606	0.586
	N (count)	123	123	123	109
Pulmonary	DNI Pearson correlation	1	−0.492	0.082	−0.054
	*p*-value (two-sided)		0.006	0.668	0.789
	N (count)	30	30	30	27
Gastrointestinal	DNI Pearson correlation	1	−0.611	0.005	0.205
	*p*-value (two-sided)		0.000	0.980	0.326
	N (count)	31	31	31	25

DNI, delta neutrophil index; MPC, mean platelet component; CRP, C-reactive protein; PCT, procalcitonin.

**Table 5 biomedicines-11-03190-t005:** Capacity of DNI and other inflammatory markers of differentiating between survival and non-survival group.

Variable	Cutoff Score	Sensitivity (%)	Specificity (%)	Positive Predictive Value (%)	Negative Predictive Value (%)
DNI (%)	6.2	100.0	80.3	10.9	100.0
Procalcitonin (mg/dL)	0.71	100.0	63.9	7.7	100.0
CRP (mg/L)	205.8	60.0	92.3	15.8	99.0

DNI, delta neutrophil index; CRP, C-reactive protein.

## Data Availability

The data presented in this study are available on request from the corresponding author.

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
