# Peer review of "Prognostic and Diagnostic Power of Delta Neutrophil Index and Mean Platelet Component in Febrile Patients with Suspected Sepsis"

_biomedicines, 2023, doi:10.3390/biomedicines11123190_

Round 1

Reviewer 1 Report

Comments and Suggestions for Authors

The authors try to demonstrate that  delta neutrophil index and mean 2 platelet component could serve as prognostic makers in febrile patients. However, only 5 patients died in the cohort, and the authors hardly focus on the mortality, but instead of identifying sepsis.  Therefore , it would be better to compare these biomarkers with old biomarkers in sepsis diagnosis, not prognosis.

The data presentation is too poor to publish. Suggest to reorganise the tables and reduced the repeating. Maximum 5 tables shall be enough.

The methodology and principle of the DNI and MPC shall be described in details . In discussion, why these markers could diagnose sepsis shall be extensively discussed.

More importantly, the sepsis diagnosis shall be described in details. 

The time of sample collection and if any longitudinal samples shall be described in details.

Comments on the Quality of English Language

need to rewrite and reorganise the manuscript 

Author Response

Response to Reviewer χ Comments

  1. Summary

Thank you very much for taking the time to review this manuscript. Please find the detailed responses below and the corresponding revisions/corrections highlighted/in tract changes in the resubmitted files.

The authors try to demonstrate that delta neutrophil index and mean 2 platelet component could serve as prognostic makers in febrile patients. However, only 5 patients died in the cohort, and the authors hardly focus on the mortality, but instead of identifying sepsis.  Therefore , it would be better to compare these biomarkers with old biomarkers in sepsis diagnosis, not prognosis.

Response -> Thank you for your Opinion. The comparison of DNI and MPC with old biomarkers in the diagnosis of sepsis mentioned by the reviewer is already well described in Tables 2 and 4. Please refer the reviewer to them.

The data presentation is too poor to publish. Suggest to reorganise the tables and reduced the repeating. Maximum 5 tables shall be enough.

Response -> Thank you for your comments, we have reorganized the tables and reduced the repetition to 5 tables as suggested by the reviewer.

The methodology and principle of the DNI and MPC shall be described in details . In discussion, why these markers could diagnose sepsis shall be extensively discussed.

Response -> Thank you for your suggestion, we have detailed the methodology and principles of DNI and MPC in the Discussion as suggested by the reviewer and discussed why these markers can diagnose sepsis. 

More importantly, the sepsis diagnosis shall be described in details. 

Response -> Thank you for your recommendation. As you mentioned, we have elaborated on the sepsis diagnosis in the discussion.

The time of sample collection and if any longitudinal samples shall be described in details.

Response -> Thank you for your recommendations. We do not have longitudinal samples of patients, and the time point of sample collection is from January 2021 to the end of March 2022, as mentioned in 2.2 study population.

Reviewer 2 Report

Comments and Suggestions for Authors

This study highlights the superiority of Delta Neutrophil Index (DNI) over other molecular biomarkers, such as procalcitonin and C-reactive protein (CRP), in differentiating febrile septic patients from non-septic febrile patients in the emergency department and predicting prognosis. The authors found MPC also contributes to the differentiation of causes in febrile sepsis. 

It is clearly presented. Just a few minor points. 

Need to introduce more background and relevant studies in the Introduction section. Discussion section is a bit long, can move the first two paragraphs from discussion to introduction.

Line 230: Previous studies (23) have shown ... Only one reference here

Line 249: Previous studies (29) have shown ...Only one reference here

No need to repeat the full name after abbreviation done already, such as line 188-190, line 258-260 and line 265. Please check the rest of the manuscript. 

Author Response

Response to Reviewer χ Comments

  1. Summary

Thank you very much for taking the time to review this manuscript. Please find the detailed responses below and the corresponding revisions/corrections highlighted/in tract changes in the resubmitted files.

Reviewer #2:

This study highlights the superiority of Delta Neutrophil Index (DNI) over other molecular biomarkers, such as procalcitonin and C-reactive protein (CRP), in differentiating febrile septic patients from non-septic febrile patients in the emergency department and predicting prognosis. The authors found MPC also contributes to the differentiation of causes in febrile sepsis. 

It is clearly presented. Just a few minor points. 

Need to introduce more background and relevant studies in the Introduction section. Discussion section is a bit long, can move the first two paragraphs from discussion to introduction.

Response -> Thank you for your opinion. We've moved the first two paragraphs of the Discussion section to the Introduction based on the reviewer's comments.

Line 230: Previous studies (23) have shown ... Only one reference here

Response -> Thank you for your opinion. In line 230, I wrote Previous studies (23) as there is only one reference, so I fixed it to previous study.

Line 249: Previous studies (29) have shown ...Only one reference here

Response -> Thank you for your suggestion. In line 249, I wrote Previous studies (29) and fixed it to previous study as there is only one reference.

No need to repeat the full name after abbreviation done already, such as line 188-190, line 258-260 and line 265. Please check the rest of the manuscript. 

Response -> Thank you for your comment. We have repeated the full names of DNI, CRP, and PCT in lines 188-190, 258-260, and 265 where we have already used the abbreviations, and these and other places in the text where we have repeated the full names, we have abbreviated them all.   

Round 2

Reviewer 1 Report

Comments and Suggestions for Authors

answered most of my questions

Comments on the Quality of English Language

minor editing is requested